# Spin-lattice couplings and effect of displacements on magnetic interactions of a skyrmion system PdFe/Ir(111)

Banasree Sadhukhan[1,2,3*], Anders Bergman[4], Johan Hellsvik[5,6], Patrik Thunström[4] and Anna Delin[3,7,8]

⋆ banasres@srmist.edu.in

**1** Department of Physics and Nanotechnology, SRM Institute of Science and Technology, Kattankulathur, 603203, Chennai, Tamil Nadu, India
**2** Tata Institute of Fundamental Research, Hyderabad, Telangana 500046, India
**3** Department of Applied Physics, School of Engineering Sciences, KTH Royal Institute of Technology, AlbaNova University Center, SE-10691 Stockholm, Sweden
**4** Department of Physics and Astronomy, Uppsala University, Box 516, SE-75120 Uppsala, Sweden
**5** PDC Center for High Performance Computing, KTH Royal Institute of Technology, SE-100 44 Stockholm, Sweden
**6** Nordita, KTH Royal Institute of Technology and Stockholm University, Hannes Alfvéns väg 12, SE-106 91 Stockholm, Sweden
**7** Swedish e-Science Research Center (SeRC), KTH Royal Institute of Technology, SE-10044 Stockholm, Sweden
**8** Wallenberg Initiative Materials Science for Sustainability (WISE), KTH Royal Institute of Technology, SE-10044 Stockholm, Sweden

## Abstract

PdFe/Ir(111) has attracted tremendous attention for next-generation spintronics devices due to existence of magnetic skyrmions with the external magnetic field. Our density functional theoretical calculations in combination with spin dynamics simulation suggest that the spin spiral phase in fcc stacked PdFe/Ir(111) flips into the skyrmion lattice phase around $B_{ext} \sim 6$ T. This leads to the microscopic understanding of the thermodynamic and kinetic behaviours affected by the intrinsic spin-lattice couplings (SLCs) in this skyrmion material for magneto-mechanical properties. Here we calculate fully relativistic SLC parameters from first principle computations and investigate the effect of SLC on dynamical magnetic interactions in skyrmion multilayers PdFe/Ir(111). The exchange interactions arising from next nearest-neighbors (NN) in this material are highly frustrated and responsible for enhancing skyrmion stability. We report the larger spin-lattice effect on both dynamical Heisenberg exchanges and Dzyaloshinskii-Moriya interactions for next NN compared to NN which is in contrast with recently observed spin-lattice effect in bulk bcc Fe and $CrI_3$ monolayer. Based on our analysis, we find that the effective measures of SLCs in fcc (hcp) stacking of PdFe/Ir(111) are $\sim 2.71(\sim 2.36)$ and $\sim 14.71(\sim 21.89)$ times stronger for NN and next NN respectively, compared to bcc Fe. The linear regime of displacement for SLC parameters is $\leq 0.02$ Å which is 0.72% of the lattice constant for PdFe/Ir(111). The microscopic understanding of SLCs provided by our current study could help in designing spintronic devices based on thermodynamic properties of skyrmion multilayers.

## 1 Introduction

In certain magnetic materials, magnetic skyrmions at the nanoscale may form. Magnetic skyrmions are stable or metastable topological magnetic textures. A magnetic skyrmion is characterized by an integer winding number or "topological charge" $Q$, defined by $Q = \frac{1}{4\pi} \int_A \vec{n} \cdot \left( \frac{\partial \vec{n}}{\partial x} \times \frac{\partial \vec{n}}{\partial y} \right) dx\,dy$, where $\vec{n}(x, y)$ is the unit vector of the local magnetization, and the integral is taken over the surface area A. Metastable magnetic skyrmions can be very long-lived and possess useful transport properties. For example, they can be manipulated using ultra-low power with help of for example the spin-orbit torque (SOT) [1]. This makes them suitable candidates for future ultra low-energy and ultrahigh-density magnetic data storage and computing applications spintronic devices like logic gates, data storage and racetrack memories [2–8]. Skyrmions are typically stabilized through the interplay of various material-specific properties such as the Heisenberg interaction, the Dzyaloshinskii–Moriya interaction (DMI), and the perpendicular magnetocrystalline anisotropy, in combination with an external magnetic field [1, 9–13].

The nanoskyrmion lattice has been reported as the magnetic ground state without external magnetic field in an ultra-thin magnetic Fe film on Ir(111), where DMI induces swirling of the spin structure [14]. Adding a non-magnetic Pd layer to Fe/Ir(111) changes the magnetic

properties, causing a spin spiral ground state, and a skyrmion lattice appears only with an applied magnetic field of a few tesla [15]. However, a microscopic understanding of why this non-magnetic surface should exhibit a complex succession of skyrmion phases has been lacking. In a recent report, it was also observed that decorating the edge of a PdFe bilayer grown on Ir(111) with ferromagnetic Co/Fe patches again induces zero-field skyrmion phase [11].

Spin is always coupled to the lattice in the topologically protected chiral spin texture of skyrmions, which leads to a local lattice distortion field [16]. The lattice acts as an additional degree of freedom to the spin degree of freedom and may even affect the magnetic skyrmion topology via the spin-lattice coupling (SLC) [16,17]. Therefore, the proper understanding of skyrmion dynamics in PdFe/Ir(111) needs the inclusion of the lattice degree of freedom. Simulating coupled spin-lattice dynamics of skyrmions at the atomistic level with first principles accuracy is much more challenging. Moving to one step down, here we used a recently developed atomistic approach of fully relativistic SLC parameters to study the effect of lattice degree of freedom in skyrmion multilayers of PdFe/Ir(111) within first principles accuracy [18,19]. However, such a simplification approach, based on the microscopic understanding of SLC parameters, can describe the qualitative interactions between the local lattice distortion due to thermal displacements and skyrmion spin texture [16]. A skyrmion-based spintronic device could be designed considering the SLC effects and will likely be very effective in precisely controlling the skyrmion for its technological applications.

In our present study, we are mostly interested in investing the SLC effect for face-centered cubic (fcc) stacking of PdFe/Ir(111) as the skyrmion phase has appeared as magnetic ground state in presence of an external magnetic field, whereas, the skyrmion phase always appears as a meta-stable state even with an external magnetic field in hexagonal close packed (hcp) stacking of PdFe/Ir(111) [20]. Here we have investigated effect of SLC on the dynamical isotropic Heisenberg exchanges and anti-symmetric DMI in fcc-PdFe/Ir(111). We computed the atomistic SLC parameters including spin-orbit coupling (SOC) from ab-initio approach for PdFe/Ir(111) considering both fcc and hcp stacking of Pd on Fe/Ir(111). The organization of the paper is as follows: In Sec. 2, we presented the necessary theoretical and computational details within the individual subsections accordingly for calculating SLC in PdFe/Ir(111) with both fcc and hcp stacking of Pd layer on Fe/Ir(111). In Sec. 3, we first investigate the magnetic ground state and its evolution with the external magnetic field in fcc-PdFe/Ir(111) from spin dynamics simulation. Then, we discuss our results on the effect of SLC parameters on the isotropic and anti-symmetric part of full magnetic interactions for fcc-PdFe/Ir(111). Next, we calculate SLC parameters for both fcc and hcp stacking of PdFe/Ir(111) and compare it with those of bulk bcc Fe [18] and CrI$_3$ monolayers [19] to get an idea of how large these effects are. Finally, in Sec. 4, we end by the conclusion and outlook.

## 2 Theoretical and computational framework

Our computational approach consists of several steps, and can be summarized as follows. Using density functional theory, we optimize the structure and compute the ground state electronic structure. The equilibrium magnetic interaction parameters are then computed using the LKAG formalism [22–27] (See Appendix C). The obtained interaction parameters are subsequently used to construct a classical spin Hamiltonian

$$\mathcal{H}_\text{S} = -\sum_{ij,\alpha\beta} J_{ij}^{\alpha\beta} S_i^\alpha S_j^\beta - \sum_i K_i^\text{U} (\mathbf{S}_i \cdot \mathbf{e}_z)^2 - \sum_i \mu_i \mathbf{B}^\text{ext} \cdot \mathbf{S}_i. \tag{1}$$

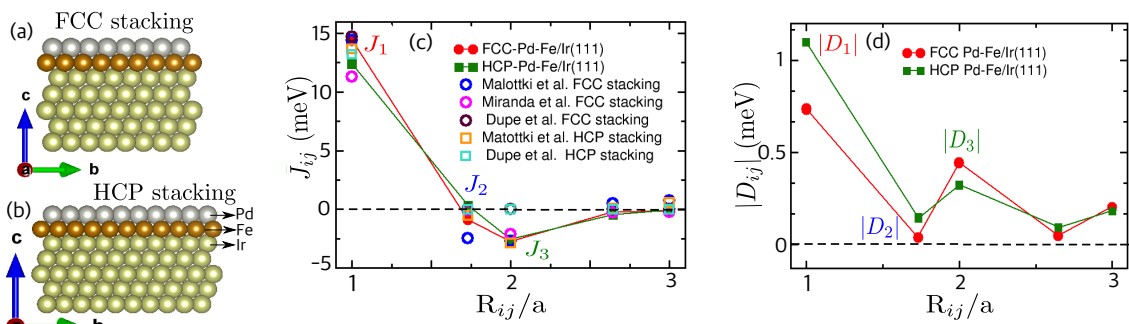

Figure 1: (a) fcc and (b) hcp stacking of PdFe bilayer on Ir(111). Calculated (c) isotropic magnetic exchange ($J_{ij}$) and (d) Dzyaloshinskii-Moriya interactions ($|D_{ij}|$) in fcc and hcp stacking of PdFe/Ir(111) with distance a is the in-plane lattice constant. Here we used the convention of positive as ferromagnetic (FM) and negative as anti-ferromagnetic (AFM) for isotropic exchange interactions. The isotropic exchange interaction ($J_{ij}$) is FM for nearest neighbour (NN) while it is AFM for the 2nd and 3rd NN. We compared our calculated $J_{ij}$ with other reports Dupe *et al.* [15], Matlottki *et al.* [20], Miranda *et al.* [21]. The skyrmion lifetimes enhances due to this frustrated exchange [20].

In the first term, we have written the bilinear exchange in the form of a tensor. The greek indices ($\alpha, \beta$) denote Cartesian coordinates ($x, y, z$), and the Latin subscripts ($i, j$) denote atomic indices. $S_i^\alpha$ is the $\alpha$-component of the normalized spin vector $\boldsymbol{S}_i$ centered on atom $i$. Note that the spins $\boldsymbol{S}_i$ are normalized to length unity in our formalism. In the second and third terms, the spins are written out explicitly as vectors. The second term is the magnetic anisotropy. Here, $\boldsymbol{e}_z$ is the easy axis unit vector, and $K_i^{\mathrm{U}}$ is the local uniaxial magnetic anisotropy at site $i$. The third term is the Zeeman term, where $\boldsymbol{B}^{\mathrm{ext}}$ is the applied external magnetic field and $\mu_i$ is the magnetic moment length at site $i$. The first term in Eq.(1) can be divided into terms describing the isotropic Heisenberg exchange (with interaction parameters $J_{ij}$), the antisymmetric exchange, i.e., the Dzyaloshinskii-Moriya interaction with parameters $\boldsymbol{D}_{ij}$, and the anisotropic symmetric exchange with parameters expressed using a symmetric matrix $\boldsymbol{C}_{ij}$, see Appendix D for details. With this spin Hamiltonian, we perform classical atomistic spin dynamics and Monte Carlo simulations (See Appendix A for details), to compute the ground state spin texture of PdFe/Ir(111) as a function of magnetic field.

To address spin-lattice coupling, we use a supercell approach, and move one atom (at a time) a small distance away from its equilibrium position. Within this new geometry, we again compute the magnetic interactions using the LKAG formalism. The spin-lattice coupling parameters (see Appendix B) can then be computed from

$$\Gamma_{ijk}^{\alpha\beta\mu} = \frac{\partial J_{ij}^{\alpha\beta}}{\partial u_k^\mu}, \tag{2}$$

where $J_{ij}^{\alpha\beta}$ denotes the bilinear exchange tensor and $u_k^\mu$ refers to the displacement of atom $k$ from its equilibrium position. Latin letters ($ijk$) represent atomic indices, and Greek letters ($\alpha\beta\mu$) represent Cartesian coordinate indices $x, y, z$. The method we use in this work to compute $\Gamma_{ijk}^{\alpha\beta\mu}$ is the same as the one in [18, 19] and in practice, a finite-difference scheme is used. A derivation and further discussion of Eq. (2) can be found in Appendix B, where we also make the connection to a rotationally invariant formalism for spin-lattice couplings. Finally, we present our results in the form of a number of scalar, averaged quantities – $J_{ij}$, $|\boldsymbol{D}_{ij}|$, $\Gamma_{ijk}$, and $|\Gamma_{ijk}|$ – defined in Appendix D.

## 2.1 Density functional calculations

Our density functional calculations for PdFe/Ir(111) have two parts. One part is the structural relaxation which has been done in calculations with the Vienna Ab initio Simulation Package (VASP) [28–30]. The other part is calculations of magnetic exchange interactions which have been done using the magnetic force theorem, as implemented in the full-potential linear muffin-tin orbital-based code RSPt [31, 32].

Magnetic exchange interactions in PdFe/Ir(111) are strongly dependent on the structural relaxation [33] which in turn affect the SLC parameters. We constructed the PdFe bilayer on a hexagonal lattice defined by five layers of the Ir(111) substrate with fcc and hcp stacking of Pd overlayer on Fe/Ir(111). The two structures are referred to as fcc PdFe/Ir(111) and hcp PdFe/Ir(111), respectively, as shown in Fig. 1(a)-(b). We put a vacuum of 20 Å along the $c$-axis. The full structural relaxations for both the stackings of PdFe/Ir(111) were done using local spin density functional (LSDA) as implemented within VASP [28–30]. The convergence with $k$-points has been carefully examined and here we used 18×18×18 $k$-point mesh with plane-wave energy cutoff 380 eV in LSDA calculations for both fcc and hcp stacking of PdFe/Ir(111). For the relaxed structures, we calculated the magnetic exchanges using LSDA with Perdew-Zunger exchange correlation functional [34] as implemented within RSPt [31, 32]. We used a 18×18×18 k-point mesh to calculate the magnetic exchanges for both the unit cell and for supercells in SLC calculations. The convergency of k-point mesh has been carefully checked by increasing the mesh size up to 24×24×18.

# 3 Results and discussions

## 3.1 Structural relaxation

To check structural relaxation, we calculated the phonon dispersion spectrum (see Fig. 7(a)-(b) in App. E). It demonstrates the absence of imaginary modes, i.e., the stabilization of the structure for both stacking. The equilibrium in-plane lattice parameter is 2.74 Å and the values of the relaxed interatomic distances are $d_{Fe-Pd} = 2.62$ Å, $d_{Fe-Ir} = 2.63$ Å respectively. We calculated the total magnetic moment of PdFe/Ir(111) is 3.39 $\mu_B$ per formula unit of PdFe/Ir(111). For Fe, we obtained a moment of 2.91 $\mu_B$/atom which induces a considerable moment in the Pd overlayer of 0.38 $\mu_B$/atom and a negligible magnetic moment of 0.06 $\mu_B$/atom in the Ir substrate respectively which are in good agreement with the reported value [15].

## 3.2 Magnetic ground state of PdFe/Ir(111)

Figure 1(c)-(d) shows the calculated isotropic part of Heisenberg exchange interactions $J_{ij}$ and DMI $\left|D_{ij}\right|$ respectively for both fcc and hcp stacking of PdFe/Ir(111). We have also performed calculations for thicker slabs using seven and nine layers of Ir, and observed that the magnetic exchanges does not change when further increasing the number of Ir layers. Both $J_{ij}$ and $\left|D_{ij}\right|$ heavily depend on the stacking sequences of the Pd layer over Fe/Ir(111). We have checked $J_{ij}$ and $\left|D_{ij}\right|$ upto seven lattice constant, but only first few nearest neighbor (NN) are contributing for PdFe/Ir(111). The first and 3rd NN of $J_{ij}$ have ferromagnetic (FM) and anti-ferromagnetic (AFM) coupling respectively for both stackings, whereas the 2nd NN has AFM and FM magnetic coupling for fcc and hcp stacking, respectively. This always leads to exchange frustration in the triangular geometry of the Fe layer (see Fig. 3(a)) which is a key issue for the stability of the skyrmion lattice phase versus the ferromagnetic phase [35].

We have investigated the magnetic ground state of fcc stacking PdFe/Ir(111) in simulations

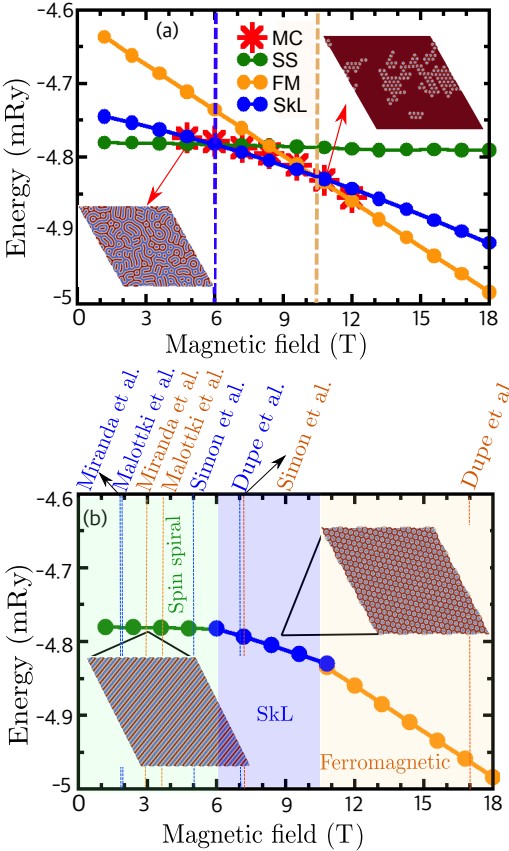

Figure 2: Data at $T = 0\,\mathrm{K}$ from spin dynamics simulations of fcc stacking of PdFe/Ir(111). (a) Energies per Fe atom of spin-spiral (SS), skyrmion lattice (SkL), and ferromagnetic (FM) spin structures in external magnetic field, obtained in spin dynamics simulation. In the Monte Carlo (MC) simulations, the initial phase has started from a random and annealed spin configuration. (b) Phase diagram with an external magnetic field. We compared our obtained phase diagram from spin dynamics simulations with other reports Dupe *et al.* [15], Matlottki *et al.* [20], Miranda *et al.* [21], Simon *et al.* [33].

of the spin Hamiltonian $\mathcal{H}_S$ for varying strength of the external magnetic field. Here we used the magnetocrystalline anisotropy energy 0.7 meV per Fe atom for fcc stacked PdFe/Ir(111) as obtained from Dupe *et al.* report [15]. Two different schemes were used for the simulations. In the first scheme, the energies as a function of external magnetic field of the particular spin structures spin spiral (SS), skyrmion lattice (SkL) and ferromagnetic state (FM) were investigated in spin dynamics simulation (SD) at zero temperature and finite damping. Here we used damping value of 0.023 as reported in the literature for PdFe/Ir(111) [21]. The external magnetic field was applied along the out-of-plane direction ($z$-axis). The energies per Fe atom are shown in Fig. 2(a) where $J_{ij}$ and $\boldsymbol{D}_{ij}$ consider upto seven lattice constant. In the second scheme the initial spin configurations were random, and annealed with heat bath Monte Carlo (MC) down to $T = 0$ K [36]. Here both the SD and MC simulations gave similar magnetic ground state as presented in Fig.2(a). For details on the employed atomistic spin dynamics and Monte Carlo methods, see Appendix A.

The energy of the SS, SkL, and FM configurations depend in different manner on the strength of the varying external magnetic field, due to the different amount of out-of-plane spin component of those spin structures which enters in the spin Hamiltonian through the Zeeman energy. The energy of the SS remains essentially constant with the external magnetic

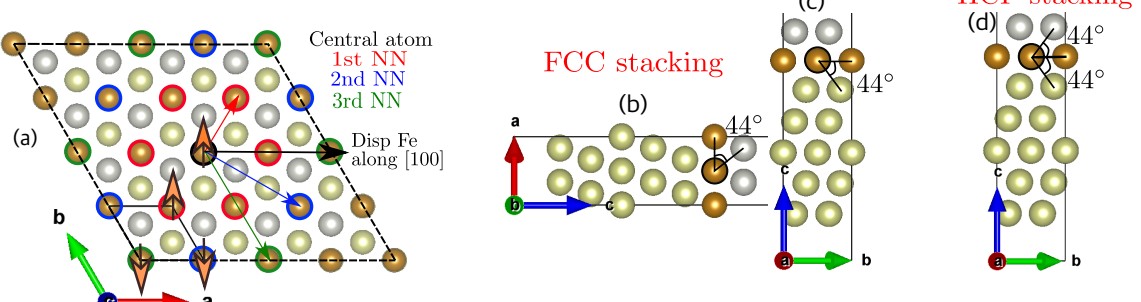

Figure 3: (a) Top view of the PdFe/Ir(111) multilayers. The filled and dotted line represent the unit cell and 4×4×1 supercell for calculating the spin-lattice coupling parameters. The atoms in different coloured circles represent the first three nearest-neighbor (NN) mapping of Fe layer. Different displacement directions in (b)-(c) fcc and (d) hcp PdFe/Ir(111). The atoms in green, brown and gray colours represent Ir, Fe, Pd respectively.

field as, the Zeeman energy $\sim 0$, whereas for the FM phase the Zeeman contribution reduces the total energy in proportion to the increase of the external magnetic field. The SkL phase also exhibits for increasing external field a decrease in energy which is related to the skyrmion density. The decrease of energy for the SkL phase is smaller compared to the case for the FM phase due to the smaller contribution of net out-of-plane component of the spins.

Figure 2(b) displays the phase diagram obtained from SD the $T = 0$ K considering $J_{ij}$ and $D_{ij}$ upto seven lattice constant. We found that the magnetic ground state without external magnetic field is a spin-spiral. Applying an external magnetic field, the SkL state lowers its energy due to the Zeeman energy, and isolated skyrmions are created in the SS background. Therefore a phase transition from SS to SkL happens at 6 T. The system undergoes another phase transition into a forced FM state for the external magnetic field 10.5 T. However, mixtures of two states such as either SS+SkL or isolated skyrmions in FM background can appear in the vicinity of the two phase boundaries in presence of an external magnetic field when we started SD and MC simulations with random and annealed spin configurations as shown in Fig. 2(a).

We found magnetic phase transitions at a larger external magnetic field than the experimental observations [37, 38], which is qualitatively consistent with the theoretical report by Dupe *et al.* [15]. Romming *et al.* reported the experimental phase transition from SS to SkL at $B_{ext} \sim 1$ T, and from SkL to FM phase at $B_{ext} \sim 2$ T with T $\sim 8$ K [38]. However, Dupe *et al.* reported SS as ground state, skyrmion above 7 T and FM above 17 T with $J_1 = 14.7$ meV. The size of the skyrmion lattice is very sensitive to the small structural modification of the thin film of PdFe/Ir(111). Simon *et al.* [33] reported that the diameter of skyrmions decrease with the relaxation of Fe layer towards inward direction of the slab which is correlated with the increasing of $|\frac{D_1}{J_1}|$ ratio. Our calculated $|\frac{D_1}{J_1}|$ ratio is 0.053 which is about to (5~6)% inward relaxation of Fe layer in PdFe/Ir(111) according to the report by Simon *et al.* [33] where the SkL appers at about 5 T and then flips into FM phase at about 7 T. This is consistent with our reported results of having different phase with external magnetic field as shown in Fig. 2(b). The diameter of a skyrmion and the smallest inter-skyrmion distance are 4.7 nm and 5.6 nm respectively which are also in very good agreement with (5~6)% inward relaxation of the Fe layer in fcc stacking of PdFe/Ir(111).

However, Miranda *et al.* reported the first phase transition from SS to SkL at about 1.8 T and the second phase transition from SkL to FM at about 3 T [21] which are 4.16 and 3.5 times smaller respectively than our simulated values. This shifting of the magnetic field to

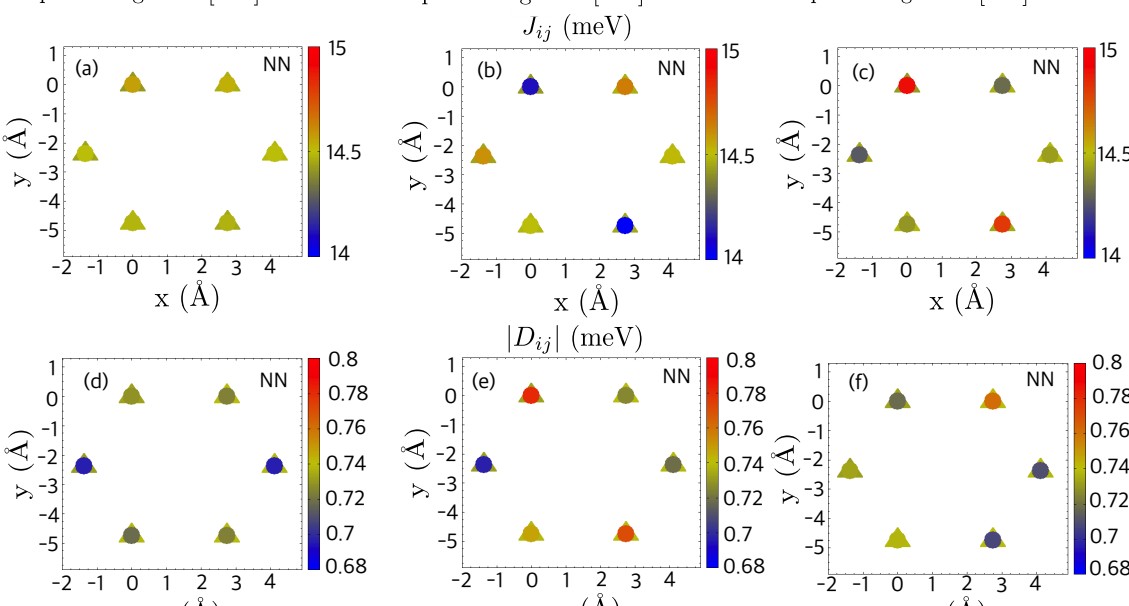

Figure 4: Calculated (a)-(c) Isotropic magnetic exchange and (d)-(f) Dzyaloshinskii-Moriya interactions in fcc PdFe/Ir(111) for 1st NN mapping with displacement (Disp) of an Fe atom along one in-plane (Fe-Fe along [100]) and two out-of-plane directions (Fe-Ir along [01$\bar{1}$] and Fe-Pd along [101] directions). The triangles and circles represent the magnetic exchanges without and with displacements respectively. The displacement amount is 0.01 Å. $J_{ij}$'s decrease with displacement of Fe atoms along Ir layers and increases with displacement of Fe atoms along Pd layer.

higher values can be interpreted in terms of isotropic part of the magnetic exchanges. The ratio of $|\frac{J_2}{J_1}|$ and $|\frac{J_3}{J_1}|$ are 0.009 and 0.188, whereas they are 0.054 and 0.196 respectively from our density functional calculations. $|\frac{J_2}{J_1}|$ is $\sim 6$ times larger compared to Miranda *et al.* report where the external magnetic field $\sim 4$ times lower value compared to our simulation. However, our calculated $|\frac{J_2}{J_1}|$ and $|\frac{J_3}{J_1}|$ are in good agreement with reported value of 0.037 and 0.21 respectively by Malottki *et al.* [20]. However, they reported the SS below 1.9 T, skyrmion between (1.7 - 3.7) T and FM above 3.7 T.

## 3.3 Choice of supercell and displacements for spin-lattice effect

The choice of supercell needed for calculations of SLC depends on the range of the magnetic interaction. As mainly the first three NN couplings are contributing to the skyrmion dynamics of PdFe/Ir(111) multilayers, the calculations of the SLC parameters $\Gamma_{ijk}^{\alpha\beta\mu}$ were done with a 2×2×1 supercell with one atom displaced to the preferred directions $u_k^{\mu}$ [18]. The convergence of magnetic exchanges are also tested with increasing the supercell size to 4×4×1. We have calculated the magnetic interactions for both the case of fcc stacking, and for the case of hcp stacking of Pd/Fe/Ir(111).

Furthermore, to check the validity of our new approach for the calculation, we checked the linear regime of SLC parameters where $\Gamma_{ijk}^{\alpha\beta\mu}$'s are independent of size of the displacements. We checked the change in isotropic part of the Heisenberg magnetic exchanges ($J_{ij}$) for different displacements amplitudes (see Fig. 6 in App. B). For displacements up to $\leq 0.02$ Å, corresponding to 0.72% of the lattice constant, the modulation of the Heisenberg exchange is linear in displacement. We identify this is a linear regime for calculation of SLC parameters.

Table 1: Extended Heisenberg parameters (in meV) without and with spin-lattice effect. Here the displacement of Fe atoms are along the out-of-plane direction ([101]). Here $J^{xx} = J^{yy} = J^{zz}$.

| NN | $J_{ij}^{xx}$ | $J_{ij}^{yx}$ | $J_{ij}^{zx}$ | $J_{ij}^{xy}$ | $J_{ij}^{yy}$ | $J_{ij}^{zy}$ | $J_{ij}^{xz}$ | $J_{ij}^{yz}$ | $J_{ij}^{zz}$ |
|---|---|---|---|---|---|---|---|---|---|
| **1st :** (Without | 14.42 | −0.28 | 0.67 | 0.28 | 14.42 | −0.031 | −0.67 | −0.031 | 14.42 |
| **2nd :** spin-lattice | −0.95 | −0.013 | 0.012 | −0.013 | −0.95 | 0.027 | −0.012 | −0.027 | −0.95 |
| **3rd :** effect) | −2.72 | 0.065 | 0.43 | −0.065 | −2.72 | −0.024 | −0.43 | −0.024 | −2.72 |
| **1st :** (With | 14.94 | 0.22 | −0.66 | −0.22 | 14.94 | −0.047 | 0.66 | −0.047 | 14.94 |
| **2nd :** spin-lattice | −1.07 | 0.04 | −0.14 | 0.04 | −1.07 | −0.09 | 0.14 | 0.09 | −1.07 |
| **3rd :** effect) | −2.76 | 0.05 | 0.27 | −0.05 | −2.76 | −0.017 | −0.27 | −0.017 | −2.76 |

The choice of the displacement directions for calculating atomistic SLC is another important aspect which depends on the crystal symmetry of the system. The in-plane motion of the atoms are considered along [100], [010] and [110] directions (see Fig. 3(a)), whereas the out-of-plane motion of atoms are along either the bare $z$ direction ($c$-axis) or along the bond length i.e Fe-Pd/Fe-Ir directions. The in-plane displacements of Fe along [100] (as shown in Fig.1(a)), [010] and [110] mean along $x$, $y$, $xy$ directions respectively. To consider a uniform displacements along the bond length towards either Pd or Ir layer, we choose the ∠Ir-Fe-Ir or ∠Pd-Fe-Pd close to $\sim 44°$ as shown in Fig. 3(b)-(d). The displacement of Fe along the Fe-Pd is [101] direction (towards Pd layer) and along the Fe-Ir is [01$\bar{1}$] direction (towards Ir layers) for fcc stacking of PdFe/Ir(111) multilayer where ∠Pd-Fe-Pd and ∠Ir-Fe-Ir are $\sim 44°$ (see Fig. 3(b)-(c)). Whereas they are [011] direction (towards Pd layer) and [01$\bar{1}$] direction (towards Ir layers) along the Fe-Pd and Fe-Ir respectively for HCP stacking of PdFe/Ir(111) multilayers for the same bond angles (see Fig. 3(d)).

## 3.4 Tailoring magnetic interactions with spin-lattice effect

Lattice vibrations always accompany magnetic excitations at finite temperatures. In coupled spin-lattice dynamics, SLC describes how lattice vibrations microscopically affect the dynamical behaviour of magnetic interactions. To investigate the microscopic effect of SLC in PdFe/Ir(111), we calculated the $J_{ij}$ and $\left|\boldsymbol{D}_{ij}\right|$ for NN with the displacement (Disp) of Fe atoms along three different directions. One is in-plane along Fe-Fe bond direction, and other two are out-of-plane along Fe-Ir and Fe-Pd bond directions respectively. Due to displacements, the monolayer of Fe in PdFe/Ir(111) loses its sixfold symmetry ($C_6$). Here we estimate an effective measure of how the dynamical Heisenberg exchanges and DMIs change with spin-lattice effect. Figure 4 presents the $J_{ij}$ and $\left|\boldsymbol{D}_{ij}\right|$ for NN for displacements of Fe atoms along Fe-Fe, Fe-Ir and Fe-Pd respectively (see Figs. 8 and 9 in App. F for the 2nd and 3rd NN, respectively). When one Fe atom in the supercell is displaced along a particular direction, some atoms come close and some atoms get away, and the magnetic exchanges change accordingly. The $J_{ij}$ changes up to $\sim 0.1$ meV for the in-plane displacement of Fe atoms, whereas it changes up to $\sim 0.5$ meV for the out-of-plane displacement of Fe atoms.

The sensitivity of magnetic exchanges is more for in-plane displacement rather than out-of-plane displacements of Fe atoms. Effective measure of the change in the $J_{ij}$ for the 1st NN is 3.46% (0.69%), where it is 12% (8%) and 1.65%(1.1%) for 2nd and 3rd NN for the out-of-plane (in-plane) displacements of Fe atoms. The extended Heisenberg parameters without and with spin-lattice effect are presented in table 1. This ensure the larger spin-lattice effect on the 2nd NN compared to 1st NN and 3rd NN for PdFe/Ir(111). The ratio of $\left|\frac{J_2}{J_1}\right|$ and $\left|\frac{J_3}{J_1}\right|$ are 0.054 and 0.196 respectively without spin-lattice effect. However, the ratio of $\left|\frac{J_2}{J_1}\right|$ and $\left|\frac{J_3}{J_1}\right|$ are

Table 2: Calculated orbital decomposed isotropic Heisenberg exchange interactions in fcc stacked PdFe/Ir(111) multilayers without displacements.

| NN | $J_{ij}^{e_g - e_g}$ (meV) | $J_{ij}^{t_{2g} - t_{2g}}$ (meV) | $J_{ij}^{t_{2g} - e_g}$ (meV) |
|---|---|---|---|
| 1st : | 5.35 | 8.85 | 0.13 |
| 2nd : | −0.41 | −0.62 | 0.11 |
| 3rd : | −0.46 | −1.72 | −0.60 |

0.071 and 0.187 for displacement of Fe atom along Fe-Fe bond length which means they are increase by $\sim 32\%$ and decrease by $\sim 4.81\%$ with the effect of SLC parameters for in-plane displacement of Fe atoms. The ratio of $|\frac{J_2}{J_1}|$ and $|\frac{J_3}{J_1}|$ are 0.0704 (0.073) and 0.183 (0.191) for displacement of Fe atom along Fe-Pd (Fe-Ir) bond length respectively.

We calculated the orbital decomposition of $J_{ij}$ in order to get an insight into the physical origin behind the observed changes in the dynamical exchange interactions $J_{ij}$ with displacements of Fe atoms. $J_{ij}$ can be presented as a sum of three contributions: $J_{ij} = J_{ij}^{e_g - e_g} + J_{ij}^{t_{2g} - t_{2g}} + J_{ij}^{t_{2g} - e_g}$ and they are presented in table 2 for fcc stacked PdFe/Ir(111) multilayers. Here the contributions of $J_{ij}^{e_g - e_g}$ and $J_{ij}^{t_{2g} - t_{2g}}$ are large compared to $J_{ij}^{t_{2g} - e_g}$ for 1st NN.

For 1st NN, all orbital contributions of magnetic exchange interactions are FM which increase slightly for in-plane displacements of Fe atoms. However, they decrease (increase) for out-of-plane displacements of Fe atoms along Fe-Ir (Fe-Pd) directions respectively. Whereas both $t_{2g} - t_{2g}$ and $e_g - e_g$ are AFM which increase (decrease) slightly due to in-plane displacements of Fe atoms for 2nd NN (3rd NN) respectively. However, $t_{2g} - e_g$ remains FM (AFM) due to displacements of Fe atom for 1st and 2nd NN (3rd NN) respectively. We also observed similarly behaviour for hcp stacking of PdFe/Ir(111) multilayers.

The $J_{ij}$ decreases whereas $\left|D_{ij}\right|$ increases for the same NN pairs for displacements of Fe atoms along Fe-Ir (see Figs. 4(b) and (e)), whereas the $J_{ij}$ increases and $\left|D_{ij}\right|$ decreases for the same NN pairs for displacements of Fe atoms along Fe-Pd (see Figs. 4(c) and (f)). The Heisenberg exchange increases for displacements of Fe atoms along Pd layer and decreases or displacements of Fe atoms along Ir layer due to band filling effects [21], whereas the DMI behaves oppositely. DMI increases for displacements of Fe atoms along Ir layer and decreases or displacements of Fe atoms along Pd layer. Effective measure of the change in the $\left|D_{ij}\right|$ are $\sim 8\%, 47\%, 15\%$ for 1st NN, 2nd NN and 3rd NN respectively for the displacement of Fe atoms along out-of-plane directions which are slightly lower in values for the in-plane motion of Fe atoms. The DMI interactions are affected more compared to Heisenberg exchanges due to spin-lattice effect. The interplay of $J_{ij}$ and $\left|D_{ij}\right|$ is an important factor for the stability of skyrmion in PdFe/Ir(111) [33].

## 3.5 Spin-lattice coupling parameters

To analyze the microscopic SLC in PdFe/Ir(111), we calculate the relativistic SLC parameters for both magnetic Fe and two non-magnetic Pd, Ir atoms as shown in Fig. 5. Here we consider three mutually orthogonal directions [100], [010] and [001] for displacement atoms to calculate SLCs. The displacement amount is 0.01 Å which is within linear regime of coupled spin-lattice dynamics. The calculated SLC parameters as a function of distance are shown in the figure 5 where the magnitude of SLC parameters gradually falls with the distance. To see the of SLC on each NN pairs, we calculated the ratio of SLC to isotropic Heisenberg exchange $|\frac{\Gamma}{J}|$ as shown in the table 3 which is larger for 2nd NN as compared 1st NN. The magnetic phase diagram with external magnetic field (described above) and lifetime of skyrmion heav-

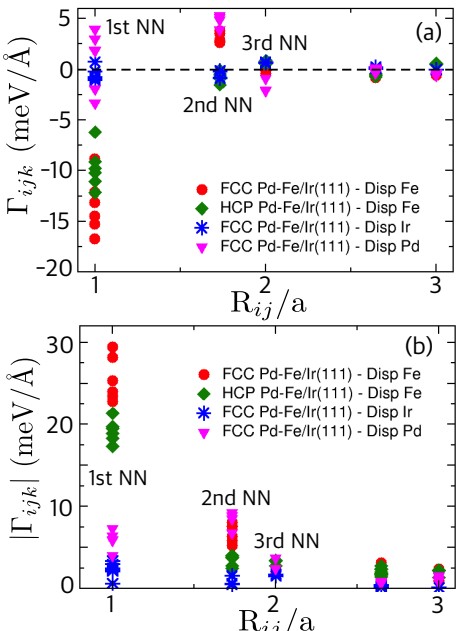

Figure 5: (a) Isotropic spin lattice coupling (SLC) parameters and (b) their absolute values for fcc and hcp stacking of PdFe/Ir(111) multilayer at different distances with displacement (Disp) of Fe, Pd, Ir atoms.

Table 3: Calculated $\left|\frac{\Gamma}{J}\right|_{avg}$ for first three NN of fcc and hcp stacking of PdFe/Ir(111) multilayers with displacement (Disp) of different atoms.

| System | $\left|\frac{\Gamma_1}{J_1}\right|_{avg}$ | $\left|\frac{\Gamma_2}{J_2}\right|_{avg}$ | $\left|\frac{\Gamma_3}{J_3}\right|_{avg}$ |
|---|---|---|---|
| fcc PdFe/Ir(111) | | | |
| Disp Fe : | 1.74 | 6.82 | 0.92 |
| Disp Pd : | 0.41 | 2.41 | 1.21 |
| Disp Ir : | 0.16 | 1.15 | 0.59 |
| hcp PdFe/Ir(111) | | | |
| Disp Fe : | 1.51 | 10.53 | 1.28 |

ily depends on the frustrated Heisenberg exchange of 2nd NN. We observed also the effects of lattice displacements on both $J_{ij}$ and $D_{ij}$ for 2nd NN are larger compared to 1st NN and 3rd NN which is consistent with the larger value of SLC parameters for 2nd NN compared to others.

To put the spin-lattice coupling in PdFe/Ir(111) in context of other magnetic materials, we compare the effective measure of SLC parameters with the exchange striction coupling of bcc Fe [18] and tne recently calculated SLC parameters in CrI$_3$ monolayer [19]. The ratio of the exchange striction coupling and the Heisenberg exchange in bcc Fe are $\left|\frac{\Gamma_1}{J_1}\right|_{avg} = 0.641\text{Å}^{-1}$ and $\left|\frac{\Gamma_2}{J_2}\right|_{avg} = 0.481$ for the 1st NN and 2nd NN pairs respectively, whereas the corresponding ratios are $\left|\frac{\Gamma_1}{J_1}\right|_{avg} = 7.42 \text{ Å}^{-1}$ and $\left|\frac{\Gamma_2}{J_2}\right|_{avg} = 2.31 \text{ Å}^{-1}$ in CrI$_3$ monolayer which are 11 and 5 times stronger than bcc Fe respectively. The ratio of the SLC parameters and the Heisenberg exchange ($\left|\frac{\Gamma}{J}\right|_{avg}$) in fcc PdFe/Ir(111) (hcp PdFe/Ir(111)) are $\sim 2.71 (\sim 2.36)$

and $\sim 14.71 (\sim 21.89)$ times stronger for 1st NN and 2nd NN respectively compared to bcc Fe. The spin-lattice effect for 2nd NN is higher compared to 1st NN in PdFe/Ir(111) which is in contrast with SLC paramters in $CrI_3$ monolayer. The ratio of the SLC parameter and the Heisenberg exchange in fcc PdFe/Ir(111) (hcp PdFe/Ir(111)) are $\sim 2.3 (\sim 4.56)$ times stronger than $CrI_3$ monolayer for 2nd NN.

## 4 Conclusion and outlook

In conclusion, at finite temperatures, spin dynamics must incorporate lattice degrees of freedom because of thermal displacements. Spin and lattice degrees of freedom are technically coupled, resulting in an intrinsic SLCs. This describes how lattice vibrations microscopically affect magnetic interaction dynamics. In our current study, we propose a framework for calculating the SLCs from first principles computation within relativistic limit for skyrmion multilayers of PdFe/Ir(111) and microscopically, examine how these couplings influence dynamical magnetic interactions in coupled spin-lattice dynamics. We predict, from a combination of first-principles calculations and atomistic spin simulations, that the magnetic ground state of fcc stacked PdFe/Ir(111) is a SS that hosts SkL with an external magnetic field $B_{ext} \geq 6$ T and enters into a forced FM phase with $B_{ext} \geq 10.5$ T. Our study suggests the existence of phase boundaries at which SS and skyrmion mix or isolated skyrmion appears in the FM background similar to experimental observations [38].

We have investigated the role of SLC on the dynamical Heisenberg exchanges and Dzyaloshinskii-Moriya interactions microscopically which are important factors for the studying the thermodynamic properties and magnetomechanical phenomena in skyrmion materials. The sensitivity of dynamical magnetic exchanges is large for out-of-plane motions of Fe atoms compared to in-plane motions and the linear regime of displacements for studying SLC parameters is $\sim 0.72\%$ of in-plane lattice constant for PdFe/Ir(111). We have calculated an effective measure of SLC parameters for both the magnetic Fe and other two non-magnetic Pd, Ir atoms in PdFe/Ir(111). The effective measures of SLCs $(\left|\frac{\Gamma}{J}\right|_{avg})$ for fcc stacking of PdFe/Ir(111) are $\sim 2.71$ and $\sim 14.71$ times stronger for NN and next NN respectively, compared to bcc Fe. However, it increases to $\sim 21.89$ times stronger for next NN compared to bcc Fe if fcc stacking changes to hcp stacking for PdFe/Ir(111) multilayers. Our current study predicts that SLC effects have great potential for precise control and stabilization of isolated skyrmions in PdFe/Ir(111) which helps in designing skyrmion-based spintronic devices depending on thermodynamic properties.

## Acknowledgments

Fruitful discussions with Yaroslav O. Kvashnin and Ivan P. Miranda are acknowledged. A.B. acknowledges eSSENCE.

**Funding information** BS acknowledges Department of Science and Technology, Government of India, for financial support with reference no DST/WISE-PDF/PM-4/2023 under WISE Post-Doctoral Fellowship programme to carry out this work. Financial support from Vetenskapsrådet (grant numbers VR 2016-05980 and VR 2019-05304), and the Knut and Alice Wallenberg foundation (grant numbers KAW 2018.0060, KAW 2021.0246, and KAW 2022.0108) is acknowledged. The computations were enabled by resources provided by the National Academic Infrastructure for Supercomputing in Sweden (NAISS) and the Swedish National Infrastructure for Computing (SNIC) at NSC and PDC, partially funded by the Swedish Research Council through grant agreements no. 2022-06725 and no. 2018-05973.

# A  Atomistic spin dynamics and Monte Carlo simulations

The (zero kelvin) ground state of the spin Hamiltonian (Eq. (1)) as a function of magnetic field was computed using atomistic spin dynamics as well as Monte Carlo simulations, as implemented in the Uppsala Atomistic Spin Dynamics (UppASD) simulation package [39]. Both methods are described briefly below. We used a cell size of 150×150×1 for both the spin dynamics and the Monte Carlo simulations.

To compute the ground state of a spin Hamiltonian within the atomistic spin dynamics approach, the atomistic Landau–Lifshitz–Gilbert (LLG) equation

$$\frac{d\boldsymbol{S}_i}{dt} = -\gamma_{\mathrm{L}}\boldsymbol{S}_i \times \boldsymbol{B}_i - \gamma_{\mathrm{L}}\alpha\boldsymbol{S}_i \times (\boldsymbol{S}_i \times \boldsymbol{B}_i)\,, \tag{A.1}$$

is evolved in time until convergence is obtained. Here, $\boldsymbol{B}_i = -\partial\mathcal{H}_{\mathrm{S}}/\partial(\mu_i\boldsymbol{S}_i)$ is the effective field on site $i$ related to the spin Hamiltonian $\mathcal{H}_{\mathrm{S}}$, in our case Eq. (1). The dimensionless (and isotropic) Gilbert damping parameter is here denoted by $\alpha$, while $\gamma_{\mathrm{L}} = \gamma/(1+\alpha^2)$ is the renormalized gyromagnetic ratio (as a function of the bare one, $\gamma$). In the presnt work, we used a time step of $\Delta t = 10^{-16}$ s to obtain the numerical solution of Eq. (A.1).

An alternative way to find the ground state of a spin Hamiltonian is to use a Monte Carlo (MC) approach. All such methods aim at calculating estimators of physical observables at a finite temperature $T$. Specifically, in this work, we have used the heat bath MC algorithm [36] to update the spin directions. The idea behind the heat bath algorithm is to assume that each spin is in contact with a heat bath and is therefore in a local equilibrium with respect to the effective field from all the other spins. Within this approach, the probability of choosing a state with energy $E$ is proportional to $\exp(-E/k_B T)$, normalized so that the total probability summed over all possible states is equal to one. Here, $k_B$ is the Boltzmann constant, and $T$ is the temperature of the system. The initial spin configurations were random, and annealed down to effectively zero kelvin.

In heat bath MC algorithm, we start from a random spin configuration, and for a given strength of the external magnetic field, the system was annealed in 10 steps from $T = 500$ K to $T = 0.0001$ K. We used $5\times10^6$ MC steps at each temperature step to thermalize the system. In the measurement phase of the MC annealing, we also used $T = 0.0001$ K and $5\times10^6$ MC steps for searching the magnetic ground state.

# B  Spin-lattice dynamics formalism: Rotationally invariant bilinear Hamiltonian

The Hamiltonian describing combined spin-lattice dynamics can be written in the general form

$$\mathcal{H}_{\mathrm{tot}} = \mathcal{H}_{\mathrm{S}} + \mathcal{H}_{\mathrm{L}} + \mathcal{H}_{\mathrm{SL}}\,, \tag{B.1}$$

where the first term, $\mathcal{H}_{\mathrm{S}}$, is a spin Hamiltonian describing purely magnetic interactions, and the second term, $\mathcal{H}_{\mathrm{L}}$, is the lattice Hamiltonian accounting for the energies associated with interatomic interactions, neglecting spin. It is well-established how to compute these two terms, see, e.g., [18].

For completeness and in order to explain our notation, we here include a brief description of the bilinear part of $\mathcal{H}_{\mathrm{S}}$ before moving on to the spin-lattice term $\mathcal{H}_{\mathrm{SL}}$. Since we are in this work already using the notation $\mathcal{H}_{\mathrm{S}}$ for a Hamiltonian containing not only bilinear terms, we will use $\mathcal{H}_{\mathrm{S}}^b$ in the following to denote the bilinear part. The spin Hamiltonian $\mathcal{H}_{\mathrm{S}}^b$ can be expressed as

$$\mathcal{H}_{\mathrm{S}}^b = -\sum_{ij,\alpha\beta} \mathcal{J}_{ij}^{\alpha\beta} S_i^\alpha S_j^\beta\,, \tag{B.2}$$

where Latin letters $(ij)$ represent atomic indices, and Greek letters $(\alpha\beta)$ represent Cartesian coordinate indices $x, y, z$. In this notation, $S_i^\alpha$ is the $\alpha$-component of the spin vector $\boldsymbol{S}_i$ centered on atom $i$. Since this is a classical spin Hamiltonian, $\boldsymbol{S}_i$ is just a three-dimensional vector in coordinate space. $\mathcal{J}_{ij}^{\alpha\beta}$ is a tensor containing the relevant spin-spin interactions, including those originating from spin-orbit coupling, e.g., the magneto-crystalline anisotropy and antisymmetric exchange, also called the Dzyaloshinskii–Moriya interaction (DMI).

The third term in Eq. (B.1), $\mathcal{H}_{\mathrm{SL}}$, i.e., the spin-lattice term, couples the spin and lattice degrees of freedom. $\mathcal{H}_{\mathrm{SL}}$ in turn consists of two ingredients – a correction term to the spin Hamiltonian due to small distortions of the atomic positions, and a correction term to the lattice Hamiltonian, due to small distortions of the spin directions. To first order we can write

$$\mathcal{H}_{\mathrm{SL}} = -\sum_{ijk,\alpha\beta\mu} \frac{\partial \mathcal{J}_{ij}^{\alpha\beta}}{\partial u_k^\mu} S_i^\alpha S_j^\beta \left(u_k^\mu - u_i^\mu\right) - \sum_{ijk,\alpha\beta\mu} \frac{\partial \Phi_{ij}^{\alpha\beta}}{\partial S_k^\mu} S_k^\alpha (u_i^\beta - u_k^\beta)(u_j^\mu - u_k^\mu). \tag{B.3}$$

Just as before, Latin letters $(ijk)$ represent atomic indices, and Greek letters $(\alpha\beta\mu)$ represent Cartesian coordinate indices $x, y, z$. $\Phi_{ij}^{\alpha\beta}$ is a tensor describing the lattice interactions, i.e., in essence, a force constant tensor. The second term in Eq. (B.3) is usually assumed to be small compared to the first term, but for some materials, the force constants may depend significantly on the spin configuration. The traditional explanation for Invar alloys relies on such a coupling between the force constants and the spin configuration [40]. Furthermore, $\boldsymbol{u}_i = \boldsymbol{x}_i - \boldsymbol{X}_i$, where $\boldsymbol{x}_i$ is the instantaneous position of atom $i$ at time $t$, and $\boldsymbol{X}_i$ is the position of the same atom $i$ at time $t = 0$. Here, we have taken the atomic positions at $t = 0$ to be their equilibrium positions. To make the functional form of the Hamiltonian rotationally invariant, its terms are expressed using differences of atomic positions in combination with time-dependent local coordinates that follow the system. In practice, the latter is realized by expressing the variables in the Hamiltonian with the help of a rotational tensor $\boldsymbol{R}(t)$.

In the present work, our focus is to understand how the isotropic Heisenberg exchange and DMI are affected when an atom is displaced from its equilibrium position. To achieve this, we compute a simplified form of $\partial \mathcal{J}_{ij}^{\alpha\beta}/\partial u_k^\mu$. Thus, in this work, we neglect the second term in Eq. (B.3), retain only the first displacement in the first term, assume that the system is not rotating, and also replace $\mathcal{J}_{ij}^{\alpha\beta}$ by $J_{ij}^{\alpha\beta}$, where $J_{ij}^{\alpha\beta}$ denotes the exchange part of $\mathcal{J}_{ij}^{\alpha\beta}$, i.e., the isotropic, symmetric anisotropic, and antisymmetric exchange terms. We use the notation

$$\Gamma_{ijk}^{\alpha\beta\mu} = \frac{\partial J_{ij}^{\alpha\beta}}{\partial u_k^\mu}, \tag{B.4}$$

for this specific part of the spin-lattice coupling (SLC). In practice, finite differences are used to compute $\Gamma_{ijk}^{\alpha\beta\mu}$, an approach that is valid if the exchange interactions change linearly with the displacement. Fig. 6, illustrates that this assumption is very reasonable for displacements $u$ up to at least 0.02 Å. We have used a displacement of 0.01 Å to calculate the SLC parameters presented in this work.

## C  LKAG formalism for calculating magnetic interaction parameters

For a given real material, the parameters above in Eq. (B.2) can be extracted from magnetic force theorem which is originally formulated for the case of isotropic Heisenberg interactions in the absence of spin-orbit coupling [22, 23]. The theory is based on linear-response theory formulated for second order perturbation in the deviations of spins from equilibrium magnetic

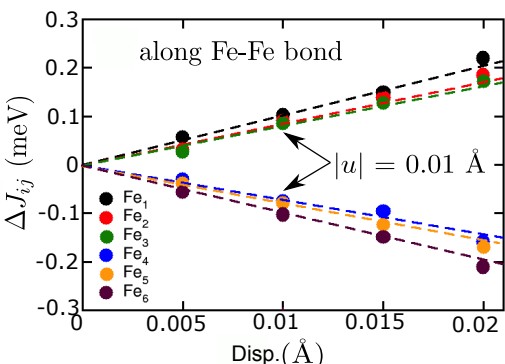

Figure 6: Isotropic exchange interaction ($J_{ij}$) as a function of displacement of the Fe atoms along the Fe-Fe bond directions for the six nearest neighbors of Fe atoms.

configuration. This perturbation in the electronic Hamiltonian is applied and the results are mapped on the classical Heisenberg model given by Eq. (B.2). The approach has been extended to take into account relativistic effects to allow to compute the full interaction tensor by several research groups [24–27,41–43].

Here we present a derivation of the formulae based on Green's functions formalism below. First we perturb the spin system by deviating the initial moments ($\vec{e}_0$) on a small angle $\vec{\delta\varphi}$ :

$$\vec{e} = \vec{e}_0 + \delta\vec{e} + \delta^2\vec{e} = \vec{e}_0 + \left[\vec{\delta\varphi} \times \vec{e}_0\right] - \frac{1}{2}\vec{e}_0(\vec{\delta\varphi})^2. \tag{C.1}$$

The Hamiltonian (Eq. (B.2)) of the perturbed system in terms of series in the order of $\vec{\delta\varphi}$ is given by:

$$\hat{\mathcal{H}}' = \hat{\mathcal{H}}^0 + \hat{\mathcal{H}}^1 + \hat{\mathcal{H}}^2. \tag{C.2}$$

In the collinear limit, all spins point along the same direction ($Z$ direction here), we can write the tilting vectors as:

$$\vec{\delta\varphi} = (\delta\varphi^x; \delta\varphi^y; 0),$$
$$\left[\vec{\delta\varphi} \times \vec{e}_0\right] = (\delta\varphi^y; -\delta\varphi^x; 0).$$

Considering the second order in $\vec{\delta\varphi}$, we can write $\hat{H}^{(2)}$ as :

$$\hat{\mathcal{H}}^2 = -\sum_{i \neq j} \left( J_{ij}^{xx} \delta\varphi_i^y \delta\varphi_j^y + J_{ij}^{yy} \delta\varphi_i^x \delta\varphi_j^x - J_{ij}^{xy} \delta\varphi_i^y \delta\varphi_j^x \right.$$
$$\left. - J_{ij}^{yx} \delta\varphi_i^x \delta\varphi_j^y - \frac{1}{2} J_{ij}^{zz} \left( (\vec{\delta\varphi}_i)^2 + (\vec{\delta\varphi}_j)^2 \right) \right).$$

Considering second order perturbation limit, the electronic Hamiltonian ($\mathcal{H}$) can be written as:

$$\hat{\mathcal{H}}' = \hat{U}^\dagger \hat{\mathcal{H}} \hat{U} = \hat{\mathcal{H}}^{(0)} + \hat{\mathcal{H}}^{(1)} + \hat{\mathcal{H}}^{(2)}, \tag{C.3}$$

where $\hat{U} = \exp(i\vec{\delta\varphi}\hat{\vec{\sigma}}/2)$ and $\hat{\vec{\sigma}}$ is the vector of Pauli matrices. Then the various components of magnetic exchange tensor $J_{ij}^{\alpha\beta}$ can be written as :

$$J_{ij}^{xx} = \frac{T}{4} \sum_n \text{Tr}_{L,m} \left[ \hat{\mathcal{H}}_i, \hat{\sigma}^y \right] G_{ij}(i\omega_n) \left[ \hat{\mathcal{H}}_j, \hat{\sigma}^y \right] G_{ji}(i\omega_n),$$

$$J_{ij}^{xy} = -\frac{T}{4} \sum_n \text{Tr}_{L,m} \left[ \hat{\mathcal{H}}_i, \hat{\sigma}^y \right] G_{ij}(i\omega_n) \left[ \hat{\mathcal{H}}_j, \hat{\sigma}^x \right] G_{ji}(i\omega_n).$$

Similarly we get other components $J_{ij}^{yy}$ and $J_{ij}^{yx}$. The summation is done over the Matsubara frequencies ($\omega_n$) where the trace is over the orbital indices denoted by $m$. The other components $J_{ij}^{xz}$, $J_{ij}^{zx}$, $J_{ij}^{yz}$, $J_{ij}^{zy}$ are not of the second order in the tilting angles ($\vec{\delta\varphi}$). Thus, for M $\parallel z$, only $D_{ij}^z$ (D.3) component can be computed, whereas $D_{ij}^x$ and $D_{ij}^y$ are extracted with the magnetization pointing along $x$ and $y$, respectively which was discussed first in Ref. [25].

# D  Definitions of the presented interaction parameter quantities

The purpose of this Appendix is to provide definitions of the quantities $J_{ij}$, $\left|\boldsymbol{D}_{ij}\right|$, $\Gamma_{ijk}$, and $\left|\Gamma_{ijk}\right|$. To define $J_{ij}$ and $\left|\boldsymbol{D}_{ij}\right|$, we decompose the bilinear exchange interaction (the first term in Eq. (1)) into three terms – an isotropic part, an antisymmetric part, and a symmetric part according to

$$\sum_{ij,\alpha\beta} J_{ij}^{\alpha\beta} S_i^\alpha S_j^\beta = \sum_{i\neq j} J_{ij} \boldsymbol{S}_i \cdot \boldsymbol{S}_j + \sum_{i\neq j} \boldsymbol{D}_{ij} \cdot \left(\boldsymbol{S}_i \times \boldsymbol{S}_j\right) + \sum_{i\neq j} \boldsymbol{S}_i \boldsymbol{C}_{ij} \boldsymbol{S}_j. \tag{D.1}$$

Here, $J_{ij}$ is the average of the diagonal components of the tensor $J_{ij}^{\alpha\beta}$, i.e.,

$$J_{ij} = \frac{1}{3}\left(J_{ij}^{xx} + J_{ij}^{yy} + J_{ij}^{zz}\right). \tag{D.2}$$

Thus, $J_{ij}$ is the usual (isotropic) Heisenberg interaction parameter. The second term is the Dzyaloshinskii-Moriya interaction term, with interaction parameters $\boldsymbol{D}_{ij}$, describing the antisymmetric part of $J_{ij}^{\alpha\beta}$. The components of the vector $\boldsymbol{D}_{ij}$ are computed from the off-diagonal components of $J_{ij}^{\alpha\beta}$. For example, the $z$-component is defined as

$$D_{ij}^z = (J_{ij}^{xy} - J_{ij}^{yx})/2. \tag{D.3}$$

$\left|\boldsymbol{D}_{ij}\right|$ is simply the length of $\boldsymbol{D}_{ij}$, i.e.,

$$\left|\boldsymbol{D}_{ij}\right| = \sqrt{(D_{ij}^x)^2 + (D_{ij}^x)^2 + (D_{ij}^x)^2}. \tag{D.4}$$

The third term in Eq. (D.1) collects the remaining parts of $J_{ij}^{\alpha\beta}$. Here, $\boldsymbol{C}_{ij}$ is a symmetric matrix. We do not analyze this term further in the present work – it is only mentioned for completeness. Finally, $\Gamma_{ijk}$, and $\left|\Gamma_{ijk}\right|$ are derived from the tensor components $\Gamma_{ijk}^{\alpha\beta\mu} = \partial J_{ij}^{\alpha\beta}/\partial u_k^\mu$ according to

$$
\begin{aligned}
\Gamma_{ijk}^\mu &= \frac{\Gamma_{ijk}^{xx\mu} + \Gamma_{ijk}^{yy\mu} + \Gamma_{ijk}^{zz\mu}}{3}, \\
\Gamma_{ijk} &= \frac{\Gamma_{ijk}^{\mu=x} + \Gamma_{ijk}^{\mu=y} + \Gamma_{ijk}^{\mu=z}}{3}, \\
|\Gamma_{ijk}^\mu| &= \sqrt{(\Gamma_{ijk}^{xx\mu})^2 + (\Gamma_{ijk}^{yy\mu})^2 + (\Gamma_{ijk}^{zz\mu})^2}, \\
|\Gamma_{ijk}| &= \frac{|\Gamma_{ijk}^{\mu=x}| + |\Gamma_{ijk}^{\mu=y}| + |\Gamma_{ijk}^{\mu=z}|}{3}.
\end{aligned}
\tag{D.5}
$$

Clearly, $\Gamma_{ijk}^{\alpha\beta\mu}$ contains many more components than the ones used in the expressions above, which are introduced for practical reasons to illustrate in a simple way the main overall effects of the spin-lattice coupling.

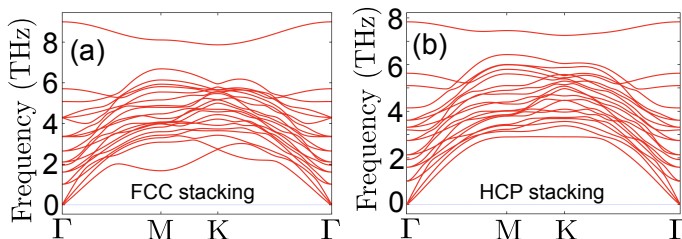

Figure 7: Phonon dispersion for (a) fcc and (b) hcp stacking of PdFe/Ir(111).

# E  Phonon dispersion in PdFe/Ir(111)

Structural relaxation was done in two steps. First, we used selective dynamics in VASP [28–30] and did ionic relaxation along the z axis of PdFe/Ir(111) multilayers. It restored the $C_6$ symmetry in the plane of Fe, Pd and Ir atoms. In the next step, we used this ionic relaxed structure for the full structural relaxation which included both ionic (for all x, y, z directions) and volume relaxations. The final relaxed structure has $C_6$ symmetry in Fe monolayer of PdFe/Ir(111). Figure 7(a)-(b) present the calculated phonon spectrum for fcc and hcp stacking of PdFe/Ir(111) respectively from finite displacement method phonon calculations with Phonopy [44] and VASP [28–30]. Absence of negative frequencies in phonon dispersion shows the the dynamical stability of PdFe/Ir(111). The full set of calculated force constants are given in table 4.

Table 4: Force constants ($\Phi_{ij}^{xx} = \Phi_{ij}^{yy} \neq \Phi_{ij}^{zz}$ for same atoms) for PdFe/Ir(111) multilayers in the unit of eV/Å$^2$.

| Atoms | $\phi_{ij}^{xx}$ | $\phi_{ij}^{xy}$ | $\phi_{ij}^{xz}$ | $\phi_{ij}^{yx}$ | $\phi_{ij}^{yy}$ | $\phi_{ij}^{yz}$ | $\phi_{ij}^{zx}$ | $\phi_{ij}^{zy}$ | $\phi_{ij}^{zz}$ |
|---|---|---|---|---|---|---|---|---|---|
| **Ir$_1$-Ir$_1$ :** | 10.42 | 0.00 | 0.00 | 0.00 | 10.42 | 0.00 | 0.00 | 0.00 | 10.97 |
| **Ir$_2$-Ir$_2$ :** | 17.09 | 0.00 | 0.00 | 0.00 | 17.09 | 0.00 | 0.00 | 0.00 | 13.89 |
| **Ir$_3$-Ir$_3$ :** | 16.39 | 0.00 | 0.00 | 0.00 | 16.39 | 0.00 | 0.00 | 0.00 | 4.05 |
| **Ir$_4$-Ir$_4$ :** | 14.87 | 0.00 | 0.00 | 0.00 | 14.87 | 0.00 | 0.00 | 0.00 | 12.17 |
| **Ir$_5$-Ir$_5$ :** | 13.56 | 0.00 | 0.00 | 0.00 | 13.56 | 0.00 | 0.00 | 0.00 | 13.77 |
| **Fe-Fe :** | 4.65 | 0.00 | 0.00 | 0.00 | 4.65 | 0.00 | 0.00 | 0.00 | 14.08 |
| **Pd-Pd :** | 10.28 | 0.00 | 0.00 | 0.00 | 10.28 | 0.00 | 0.00 | 0.00 | 6.14 |
| **Fe-Pd :** | 0.43 | 0.23 | 1.25 | 0.23 | 0.16 | 0.72 | 0.66 | 0.38 | 1.69 |
| **Ir$_1$-Fe :** | 0.02 | 0.01 | 0.04 | 0.01 | 0.02 | 0.02 | 0.01 | 0.00 | 0.04 |
| **Ir$_1$-Pd :** | 0.01 | 0.00 | 0.00 | 0.00 | 0.01 | 0.00 | 0.00 | 0.00 | 0.02 |
| **Ir$_2$-Fe :** | 0.02 | 0.00 | 0.00 | 0.00 | 0.00 | 0.01 | 0.00 | 0.01 | 0.01 |
| **Ir$_2$-Pd :** | 0.00 | 0.00 | 0.01 | 0.00 | 0.00 | 0.00 | 0.01 | 0.01 | 0.01 |
| **Ir$_3$-Fe :** | 0.00 | 0.00 | 0.00 | 0.00 | 0.00 | 0.00 | 0.00 | 0.00 | 0.00 |
| **Ir$_3$-Pd :** | 0.00 | 0.00 | 0.00 | 0.00 | 0.00 | 0.00 | 0.00 | 0.00 | 0.00 |
| **Ir$_4$-Fe :** | 0.01 | 0.04 | 0.05 | 0.04 | 0.06 | 0.03 | 0.12 | 0.07 | 0.31 |
| **Ir$_4$-Pd :** | 0.02 | 0.00 | 0.00 | 0.00 | 0.02 | 0.00 | 0.00 | 0.00 | 0.00 |
| **Ir$_5$-Fe :** | 0.11 | 0.00 | 0.00 | 0.00 | 1.42 | 2.21 | 0.00 | 2.29 | 3.05 |
| **Ir$_5$-Pd :** | 0.00 | 0.01 | 0.02 | 0.01 | 0.01 | 0.01 | 0.03 | 0.02 | 0.11 |

# F  Next nearest neighbor spin-lattice couplings

Figures 8 and 9 present calculated isotropic magnetic exchange and Dzyaloshinskii-Moriya interactions for 2nd NN (next NN) and 3rd NN respectively. Here the displacements of Fe atom are chosen for one in-plane direction (along Fe-Fe bonds) and two out-of-plane directions (along Fe-Ir and Fe-Pd bonds) respectively. The displacement amplitude of 0.01 Å is within the linear regime of coupled spin-lattice dynamics.

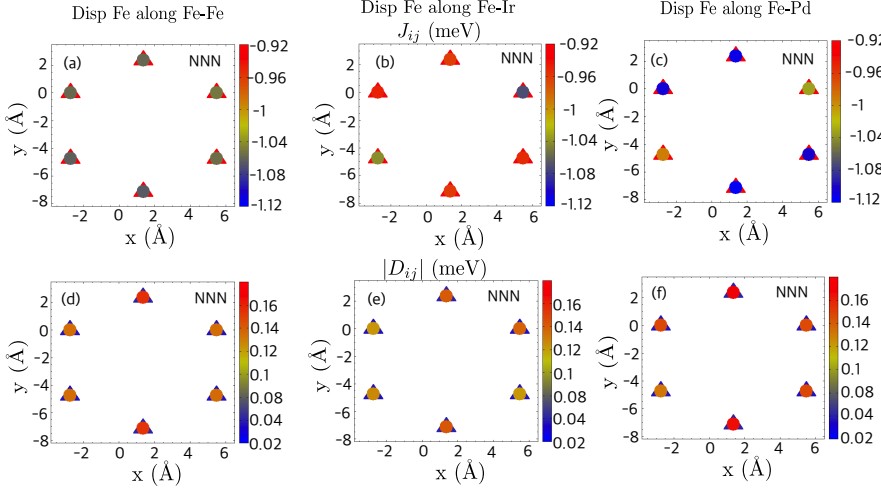

Figure 8: Calculated (a)-(c) isotropic magnetic exchange and (d)-(f) Dzyaloshinskii-Moriya interactions in fcc PdFe/Ir(111) for next nearest-neighbor (NNN) or 2nd NN mapping with displacement (Disp) of Fe atom along one in-plane (Fe-Fe along $[100]$) and two out-of-plane directions (Fe-Ir along $[01\bar{1}]$ and Fe-Pd along $[101]$ directions). The triangles and circles represent the magnetic exchanges without and with displacements respectively. Here we used displacements of 0.01 Å.

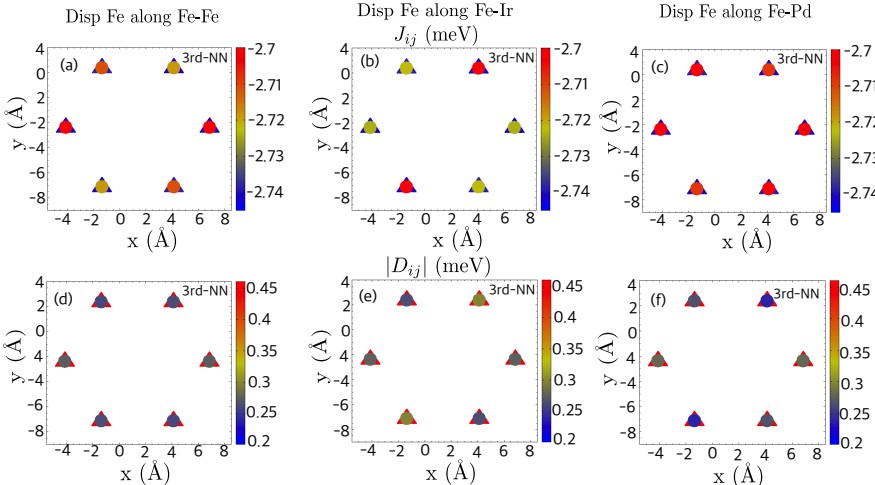

Figure 9: Calculated (a)-(c) isotropic magnetic exchange and (d)-(f) Dzyaloshinskii-Moriya interactions in fcc PdFe/Ir(111) for 3rd nearest-neighbor (NN) mapping with displacement (Disp) of Fe atom along one in-plane (Fe-Fe along $[100]$) and two out-of-plane directions (Fe-Ir along $[01\bar{1}]$ and Fe-Pd along $[101]$ directions). The triangles and circles represent the magnetic exchanges without and with displacements respectively. Here we used displacements of 0.01 Å.

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
