# Peer review of "Spin-lattice couplings and effect of displacements on magnetic interactions of a skyrmion system PdFe/Ir(111)"

_SciPost Physics, doi:SciPost Phys. 18, 064 (2025)_

## Round 1 · Referee Report · Anonymous (Referee 1) · 2024-10-15

Strengths

1- very original. The coupling between phonon and magnetic interactions is usually not studied because it very complex. Here the authors have used a combination of codes (VASP, Phonopy and home made second principle code) and techniques (MC, DFT) and are able to explain very well the outcome. 2- excellent bibliography. Pd/Fe/Ir(111) is a prototypical skyrmion ultra-thin film. There are a lot of work in this subject and it is sometimes difficult to navigate in all the references. The authors have done a great job in this respect. 3- clearly written

Weaknesses

1- the figures. Please put the article references on fig.1 and fig.2 Use thicker dashed lines in panel b like in panel a. Use also smaller symbols on panel a 2- the phonon spectra is not really useful here. It is mentioned that it is only here to check the stability of the ultra thin film however, the stability of PdFe/Ir(111) is not a key finding. It would be more interesting to study the substitution and intermixing. With the phonon band structure, one can do more that what is done in the paper i.e. establishing an elastic model for thermal phonon and coupling it to the extended Heisenberg. Since all phonon were calculated, the displacement can also be done along a specific phonon - also not done. 3- the coupling between u and J or D is also not mentioned - the model could be refined here. 4- some typos (appendix C - formaliism) 5- the effects of the displacement on the phase diagrams are not

Report

The article "Microscopic effect of spin-lattice couplings on dynamical magnetic interactions of a skyrmion system PdFe/Ir(111)" by Sadhukhan et al reports on the effect of atomic displacement on the parametrization of an extended Heisenberg Hamiltonian. PdFe/Ir(111) is one of the protypical skyrmion system. This system has been studied extensively since the last ten years and is now a very good test case. It is now being used to explore the coupling between magnetism and phonon displacement. This is an interesting topic which could have implications in the future spintronics and skyrmionics where the stability of skyrmions with respect to temperature is a crucial quantity and is a directly link to thermal phonon.

The article is well written and the figure are appropriate but could be improved.

Appart from the comment above, I have a two requests. - The effect of the phonon is explored but no table summarizes the Hamiltonian parameters is given in the paper. It would be interesting to have these quantities to explore the stability diagram depending on phonon displacement. - the force constants obtained from the band structure should be published. As I said before the stability of this ultra thin film is not really interesting because it is obviously stable. The elastic parameters are more important. Even if the authors do not have access to a code with phonon and Heisenberg parameters, other people in the community might be interested by this quantity.

Requested changes

1- table with the different parameters of the extended Heisenberg

Recommendation

Ask for minor revision

  • validity: top
  • significance: top
  • originality: top
  • clarity: high
  • formatting: excellent
  • grammar: excellent

Author:  Banasree Sadhukhan  on 2024-12-03  [id 5021]

(in reply to Report 1 on 2024-10-15)
Category:
answer to question
correction

Please find the attached referee response.

Attachment:

Referee-response-SciPost.pdf

---

## Round 1 · Referee Report · Anonymous (Referee 2) · 2024-11-21

Strengths

0- The manuscript gives a detailed investigation of spin-lattice couplings (SLC) in a system where a skyrmion lattice can be induced by a magnetic field.

1- In principle, this could now be the starting point to check how structural modifications (due to anharmonic effects at finite temperature or at step-edges) change the stability of the skyrmion lattice.

2- Large SLC probably means also that the structure is modified by magnetic pattern. Maybe the authors can at some point estimate these changes for a skyrmion lattice of given size (2nd term in eq. B3).

Weaknesses

1- In the title, 'dynamical magnetic interactions' are mentioned, so I expected to see some impact of SLC on the spin-dynamics. Maybe 'magnetic interactions' would be more clear.

2- See list of requested changes below.

Report

  1. The system Pd/Fe/Ir(111) was one of the first, where a skyrmion lattice could be induced via an external magnetic field. The stability depends sensitively on the stacking, and small structural changes can probably have a large effect on the exchange parameters. Here, the authors investigate the dependence of Heisenberg parameters (J) and the Dzyaloshinskii-Moriya interaction (DMI) on small displacements of the Fe positions, termed spin-lattice coupling (SLC). The parameters (up to 3rd nearest neighbors) are evaluated with density functional theory (DFT) and the stability of the skyrmion lattice checked with atomistic spin-dynamics simulations. Compared to systems investigated by some of the authors in other papers (CrI3 and bcc Fe), large valued of the SLC were found.

  2. The manuscript is in most parts clearly written and addresses the topic in depth, there are, however, a few points that should be clarified (see below).

Requested changes

1- The interlayer distances given on page 3 are too large. In an fcc(111) film, the ratio between interlayer distance and in-plane lattice constant should be sqrt(2/3). For Pd/Fe and Fe/Ir one expects them to be even smaller (see ref.[15]).

2- The type of exchange-correlation functional should be specified. Probably there are many different LSDA's in the used DFT codes, e.g. Perdew-Zunger or Vosko-Wilk-Nussair.

3- Specifying the displacements, the authors use directions like [100] or [101]. Probably this does not correspond to the Ir lattice, please introduce the notation or use the bulk Ir directions as reference.

4- Related to that, looking at Fig.4, I did not figure out in which direction the Fe was displaced. Assuming that the shift in (b) and (e) is in y-direction, I had expected a (vertical) mirror symmetry in the panels. Maybe the authors can indicate the shift direction in the panels.

5- In the introduction, the authors write that 'A is the entire plane perpendicular to the propagation direction'. I don't know what propagates here.

6- On page 6 I read that 'due to thermal displacements' the Fe layer loses its 6-fold symmetry. What are thermal displacements in this connection? Anharmonic vibrations should mainly occur along the surface normal. In-plane, the layer lost it 6-fold symmetry already due to the substrate and overlayer.

7- In the references, please correct the capitalization in the titles (e.g. CrI3 instead of cri3).

Recommendation

Ask for major revision

  • validity: high
  • significance: high
  • originality: high
  • clarity: good
  • formatting: perfect
  • grammar: excellent

Author:  Banasree Sadhukhan  on 2024-12-05  [id 5026]

(in reply to Report 2 on 2024-11-21)
Category:
answer to question
correction

Please find the attached referee response.

Attachment:

Referee-response-Scipost.pdf

---

## Round 2 · Referee Report · Anonymous (Referee 1) · 2025-1-3

Report

The authors partially modified the manuscript according to the referee reports, however there are a few open points (see below).

Requested changes

1- On page 2, they call "u" now a "thermal displacement". This displacement might be due to thermal motion, but it could also result from other sources, e.g., an electric field. At the moment, it is just a displacement and should be called so.

2- New table I: since J^xx = J^yy = J^zz and J^xy=-J^yx etc. only four out of nine tensor components need to be shown (or J and D_x, D_y, D_z).

3- A force constant is given in Appendix E, but it's not clear for which atoms and why the xx, yy, and zz components are identical in a multilayer film.

4- About the title: In an experiment at finite temperatures, magnetic interactions are naturally modified by the dynamics of the system, sure. What is calculated here (eq.2) is the effect of a displacement on a magnetic interaction at T=0, therefore, I don't think that "spin-lattice couplings on dynamical magnetic interactions" is a good description of the content.

5 - About the interlayer distances: since the system is stable and the exchange interactions agree also with other references, I think the results are OK, but maybe the authors give interatomic distances instead of interlayer distances.

6 - A reference for the Perdew-Zunger LSDA is missing.

7- About fig.4: Indeed, panels (b) and (c) show mirror symmetry, but along a plane 60 degrees tilted from the y-axis (for both panels). How does this match with the (in both panels, different) displacement directions?

8- About the 6-fold symmetry: of course, a free monolayer has C_6 symmetry, but the "monolayer of Fe in PdFe/Ir(111)" has not, irrespective of thermal displacements.

Recommendation

Ask for minor revision

  • validity: high
  • significance: high
  • originality: high
  • clarity: good
  • formatting: excellent
  • grammar: excellent

Author:  Banasree Sadhukhan  on 2025-01-16  [id 5125]

(in reply to Report 1 on 2025-01-03)
Category:
answer to question
correction

Please find the attachment.

Attachment:

Response-Second.pdf

---

## Editorial Decision

published